# Lanthanum Significantly Contributes to the Growth of the Fine Roots’ Morphology and Phosphorus Uptake Efficiency by Increasing the Yield and Quality of *Glycyrrhiza uralensis* Taproots

**DOI:** 10.3390/plants13040474

**Published:** 2024-02-07

**Authors:** Tingting Jia, Junjun Gu, Miao Ma, Yuyang Song

**Affiliations:** 1Ministry of Education Key Laboratory of Xinjiang Phytomedicine Resource Utilization, College of Life Sciences, Shihezi University, Shihezi 832003, China; jiatingting@stu.shzu.edu.cn (T.J.); 20202006035@stu.shzu.edu.cn (J.G.); 2Agriculture College, Shihezi University, Shihezi 832003, China

**Keywords:** *Glycyrrhiza uralensis*, phosphorus uptake, secondary metabolism, phenotypic plasticity of root system, LaCl_3_

## Abstract

The occurrence of different degrees of phosphorus deficiency in the vast majority of *G. uralensis* cultivation regions worldwide is common. There is a pressing need within the cultivated *G. uralensis* industry to identify appropriate exogenous substances that can enhance the uptake of phosphorus and improve both the yield and quality of the taproots of *G. uralensis*. This study was conducted to investigate the fine root and taproot morphology, physiological characteristics, and secondary metabolite accumulation in response to the supply of varying concentrations of LaCl_3_ to *G. uralensis*, to determine the optimal concentration of LaCl_3_ that can effectively enhance the yield and quality of *G. uralensis*’s taproots, while also alleviating its reliance on soil phosphate fertilizer. The findings indicate that the foliar application of lanthanum enhanced root activity and increased APase activity, eliciting alterations in the fine root morphology, leading to promoting the accumulation of biomass in grown *G. uralensis* when subjected to P-deficient conditions. Furthermore, it was observed that the nutrient uptake of *G. uralensis* was significantly improved when subjected to P-deficient conditions but treated with LaCl_3_. Additionally, the yield and quality of the medicinal organs of *G. uralensis* were significantly enhanced.

## 1. Introduction

The 20th century has witnessed an increasing escalation in serious agricultural production challenges on a global scale [1,2]. The scientific community has recently focused on researching the incorporation of safe exogenous chemicals into agricultural practices. This interest stems from the increasing importance placed on sustainable agriculture and environmental conservation. The aim is to identify compounds that can enhance crop growth and development, while simultaneously lowering the reliance on conventional soil chemical fertilizers [3,4].

*G. uralensis* (*Glycyrrhiza uralensis* Fisch.) is a perennial herbaceous plant of the genus *Glycyrrhiza* linn of the family *Leguminosae*, which is extensively cultivated in China [5]. *G. uralensis* is widely recognized as one of the most commercially valuable species in the global medicinal plant market [6]. The taproots, serving as the primary medicinal organs, exhibit a diverse array of applications and have substantial demand within traditional Chinese medicine, healthcare products, cosmetics, the food industry, and pharmaceutical production [7,8,9,10]. In clinical applications inside China, the utilization of *G. uralensis* is frequently observed in the form of medicinal slices, which are extensively utilized. The quality of *G. uralensis* medicinal slices mainly relies on the morphology of the main roots, which is characterized by thickness and strength, as well as their high concentration of medicinal ingredients [11].

Over the span of the last five decades, there has been a consistent upward trend in the demand for *G. uralensis*, accompanied by a corresponding increase in its consumption. The frequent occurrence of the overexploitation of *G. uralensis*, motivated by the substantial economic benefits, has led to the ongoing depletion of wild *G. uralensis* resources in China [12,13]. To reconcile the problem posed by the scarcity of wild *G. uralensis* resources, cultivated *G. uralensis* has progressively become a substitute for wild *G. uralensis*. Despite the increasing cultivation of *G. uralensis*, there is a lack of consistency in the quality of their taproots and relatively low production [14]. This has emerged as a significant constraint in the sustainable development of the *G. uralensis* industry. Consequently, enhancing the yield and quality of cultivated *G. uralensis* has become a central concern for the efficient progress of the industry.

Phosphorus is one of the essential elements in facilitating the growth and development of plants [15]. Nevertheless, the quantity and speed at which phosphorus-containing compounds are replenished in the natural environment are limited, and the compound exhibits properties such as limited solubility, restricted mobility, and substantial fixation, especially with limited P soil solubility, which restricts P uptake by higher plants [16]. Consequently, terrestrial ecosystems globally exhibit varying levels of “phosphorus limitation” [17,18], with around 70% of soils in China experiencing phosphorus deficit [19].

Xinjiang is the primary cultivation region of *G. uralensis* in China [5]. The arid and hot climatic conditions, coupled with soil characteristics such as coarse sand and substantial salt content, contribute to the persistent low levels of available phosphorus in the soils of *G. uralensis* cultivation areas in this region [20]. This is the key reason for relying on the application of large quantities of phosphorus-containing fertilizers in cultivated licorice in Xinjiang. The excessive application of chemical fertilizers ultimately leads to detrimental effects on soil structure and function, resulting in decreased crop yield and quality [21], which significantly hampers the potential for sustained development within the *G. uralensis* industry. Hence, the investigation of novel approaches to enhance phosphorus use efficiency holds significant scientific merit for developing farmland management measures that are conducive to environmental protection and the sustainable, healthy development of the *G. uralensis* industry.

Lanthanum (La, III) is a rare earth element with abundant reserves in China. Due to its distinctive physicochemical characteristics, lanthanum has found use not only in conventional industrial and medical sectors [22,23], but also in the realm of agricultural production [24]. The presence of appropriate quantities of lanthanides has been demonstrated to have a positive effect on plant growth and the synthesis and accumulation of secondary metabolites. However, the vast majority of research conducted thus far is concentrated on the effects of lanthanides on the enhancement of crop yield and quality in aboveground plant parts, specifically in crops such as tea (*Camellia sinensis* L.) [25], rice (*Oryza sativa* L.) [26], wheat (*Triticum aestivum* L.) [27], maize (*Zea mays* L.) [28,29], and legumes (*Vigna angularis* L., *Glycine max* L.) [30,31]. Lanthanides, in actuality, typically accumulate predominantly in the root systems of plants. These underground portions are more susceptible than the aboveground parts to the impacts of lanthanides due to the underground portions’ high plasticity and sensitivity. However, little research has been undertaken to investigate the effect of lanthanides on the growth and development of crop root systems [32,33].

Hence, our research primarily investigated the primary roots, which serve as the location for the biosynthesis and accumulation of secondary metabolites, as well as the fine roots, which play a crucial role in the absorption of water and minerals, of cultivated *G. uralensis*. We investigated the effects of foliar spraying of LaCl_3_ on the yield and quality of *G. uralensis* taproots. Additionally, we analyzed the influence of LaCl_3_ on the absorption of available phosphorus and phosphorus redistribution within the root system. Furthermore, we evaluated the safety of *G. uralensis* taproots after the addition of LaCl_3_. The objective of this study was to determine the optimal concentration of LaCl_3_ that can effectively improve the yield and quality of the taproots of *G. uralensis*, and the findings of this research will contribute to the creation of scientifically based cultivation measures aimed at achieving high yields and superior quality of *G. uralensis*.

## 2. Materials and Methods

### 2.1. Test Materials

The seeds of *G. uralensis* utilized for the experiment were provided by the Institute of Licorice at Shihezi University (Shihezi, China). The standards of glycyrrhizic acid (Batch No.: C11654946), glycyrrhizin (Batch No.: C11602211), glycyrrhizin (Batch No.: C10828731), methanol, formic acid, anhydrous ethanol, 2,3,5-triphenyl tetrazolium chloride (Batch No.: C298-96-4), ethyl acetate, KH_2_PO_4_, 2,4-dinitrophenol indicator, NaOH, ammonium vanadate, aluminum trichloride, and sulfuric acid were purchased from Macklin Biochemical (Shanghai Macklin Biochemical Co., Ltd., Shanghai, China); acetonitrile was purchased from Thermo Fisher Scientific (LCMS-level, Thermo Fisher Scientific Co., Ltd., Waltham, MA, USA); nitric acid and hydrogen peroxide were purchased from Adamus Reagent (Adamas Reagent Co., Ltd., Shanghai, China).

### 2.2. Plant Growth Conditions

The experimental site was situated in an open area on the Ministry of Education Key Laboratory of Xinjiang Phytomedicine Resource Utilization at Shihezi University campus (44°31′10″ N, 86°06′94″ E), at an altitude of about 450 m above sea level, with a temperate continental climate. The recorded climate data of the growth period of *G. uralensis* in 2020, including the average monthly maximum and minimum temperatures, cumulative monthly precipitation, and average monthly duration of sunlight, are listed below: May (15–25 °C, 18 mm, and 14 h 45 min), June (20–30 °C, 22 mm, and 15 h 25 min), July (22–32 °C, 22 mm, and 15 h 6 min), August (20–31 °C, 19 mm, and 14 h 39 min), August (20–31 °C, 19 mm, and 14 h 39 min) and September (14–25 °C, 13 mm, and 12 h 49 min). Temperatures and sunshine duration data were collected from the China Meteorological Science Data Sharing Service System (https://data.cma.cn/, accessed on 1 January 2021), and precipitation data were collected from the National Aeronautics and Space Administration’s (NASA) MERRA-2 Modern Retrospective Analysis (https://gmao.gsfc.nasa.gov/reanalysis/MERRA-2/, accessed on 1 January 2021).

### 2.3. Cultivation Substrate Collection

The cultivation substrate utilized in this study imitated the characteristic soil composition found in the *G. uralensis* cultivation region, which predominantly consists of sandy soil (sand:loam mixture ratio of 3:7). The sand was sourced from the southern periphery of the Gurbantunggut desert, while the loam was collected from the experimental site of Shihezi University. Subsequently, the two materials were thoroughly blended, then the mixture was separated evenly into 48 plastic pots (diameter: 28 cm; height: 30 cm) with an average weight of 9 kg for each one.

The physicochemical characteristics of the mixed cultivation substrate were as follows: total nitrogen content was 0.268 g/kg, total phosphorus content was 0.0855 g/kg, total potassium content was 5.72 g/kg, available nitrogen content was 43.59 mg/kg, available phosphorus content was 4.1 mg/kg, available potassium content was 119.09 mg/kg, and organic matter content was 5.81 g/kg. The extraction method, standard curve, and determination were referenced from Tang [34] and Sikder [35].

### 2.4. Seed Pre-Treatment

Healthy, uniformly sized, and mature *G. uralensis* seeds were subjected to a soaking process with a concentration of 98% sulfuric acid solution for 0.5 h. Then, the seeds were thoroughly rinsed under running water to ensure the absence of any residual sulfuric acid on their surface. Subsequently, the seeds were immersed in distilled water at a temperature of 25 °C for eight hours.

### 2.5. Experimental Design

Two phosphorus supply levels were set up in this experiment: (1) P-sufficient conditions: phosphorus supplementation was conducted in accordance with the fertilization pattern of *G. uralensis* farmland. The KH_2_PO_4_ was uniformly blended into the soil in pots, according to an application amount of 0.05 kg/m^2^; (2) P-deficient conditions: phosphorus fertilizer application was half of the P-sufficient conditions. The KH_2_PO_4_ was also uniformly blended into the soil in pots, according to an application amount of 0.025 kg/m^2^. After fully mixing, the soil available phosphorus concentrations were measured to be 51.2 mg/kg and 23.2 mg/kg, respectively. The determination method was referenced from the China National Standard (HJ 704-2014) [36]. Twenty-four repetitions were set for each treatment.

On 18 May 2020, the seeds with complete imbibition were selected, and six *G. uralensis* seeds were evenly sown in the cultivation substrate of each plastic pot at a depth of 1 cm. The setting of the planting density in this study was based on the initial planting density obtained from licorice farms (6.5–7.0 × 10^5^ plants/hm^2^). The final planting density of four plants per pot was achieved by the calculation of the plastic pot’s area. As the third true leaves emerged, two excess seedlings were removed according to the final planting density, leaving four uniformly growing seedlings per pot.

After 10 days, fertilizers were applied to each pot based on the average level of the *G. uralensis* field, with a mixture of 14.99 g/m^2^ of urea (N ≥ 46%) and 10.49 g/m^2^ of potassium sulfate (K_2_O ≥ 50%), and it was applied five times, once every 25 days.

Four LaCl_3_ solution concentrations were set up in P-sufficient and P-deficient cultivation substrates, respectively: 0 mM, 0.2 mM, 0.4 mM and 0.6 mM. Upon the complete expansion of the fifth true leaves, spray bottles were used to uniformly spray different concentrations of LaCl_3_ solution on the foliage of *G. uralensis* seedlings, and we applied an equal amount of distilled water to the control check (0 mM). During this experiment, we ensured that the leaf adaxial and abaxial surfaces were sufficiently moistened during spraying, which was repeated weekly for a total of nine sprays. Each treatment was repeated six times.

All pots were randomly placed in an open area on the Ministry of Education Key Laboratory of Xinjiang Phytomedicine Resource Utilization at Shihezi University campus with a distance of 50 cm between two adjacent pots to avoid shading among the treatments.

They were also organized in a randomized block group pattern, with the placements being randomly swapped every week, to eliminate the influence of light and marginal effect. The experiment included daily watering, utilizing the weighing method to maintain the soil’s moisture at 80% of its water-holding capacity.

### 2.6. Measurement of Morphological Parameters of the Root System

The experimental materials were collected on 6 September 2020, the aboveground organs of *G. uralensis* were cut off from the pot, and the whole soil and root systems were cautiously extracted from the plastic pots, ensuring the roots remained undamaged. Subsequently, the root systems were immersed in water and rinsed gently to remove all soil particles adhering to them, and the rhizomes were separated from the roots.

The cleaned roots were placed in a transparent plastic tray containing water; this arrangement allowed the fine roots to be evenly distributed in the water for imaging. The scanning process was conducted with an Epson digital scanner (Expression 11000XL; Epson, Suwa City, Japan). The root Image Analysis System software (WinRHIZO Pro 2013a; Regent Instruments Inc., Quebec City, QC, Canada) was utilized to measure various root parameters including total root length (TRL), taproot length (TL), maximum diameter of taproots (TD), fine root length (FRL), fine root diameter (FRD), and fine root surface area (FRA).

### 2.7. Determination of RA and Fine Root APase Activity

The 2,3,5-triphenyltetrazolium chloride (TTC) method [37] was used to determine the activity of roots (RA) in this experiment, and TTC reductive intensity was used to represent root activity.

Initially, 0.300 g of fine root tissue of each sample was accurately weighed and transferred into individual beakers, then each beaker was filled with 5 mL of pre-prepared 0.4% TTC solution and 5 mL of 0.1 M phosphate buffer (pH = 7.0) and maintained in the dark at 25 °C for two hours. Then, 2 mL of 1 M sulfuric acid was added to terminate the reaction. Subsequently, root tissue was removed from beakers and individually transferred to mortars after thorough rinsing. In each mortar, a total volume of 4 mL of ethyl acetate was added together with a minor quantity of quartz sand; the additions were to facilitate the complete grinding and extraction of tribenzohydrazone. After that, the crimson extract was placed into test tubes, and ethyl acetate was added for a final volume of 10 mL. The absorbance measurement at a wavelength of 485 nm was conducted with a UV–visible spectrophotometer (UV-1900, Shimadzu Corporation, Shanghai, China), and the formula was as follows:TTC reductive intensity (mg/(g·h)) = m/m_0_·t(1)
m denotes the amount of TTC reduction (mg); m_0_ denotes the fresh weight of the sample of roots (g); t denotes the reaction time (h).

The determination and standard curve for the root acid phosphatase (APase) activity of *G. uralensis* were referenced from the study conducted by McLachlan et al. [38]. The absorbance measurement at a wavelength of 410 nm was conducted with a UV–visible spectrophotometer (UV-1900, Shimadzu Corporation, Shanghai, China), and the formula was as follows:root APase activity (μg/h/g) = n × M × dilution/(W × t)(2)
n denotes the concentration of p-nitrophenol in the corresponding sample (μmol) and M denotes the relative molecular mass of p-nitrophenol (139.11 g/mol). W denotes the fresh weight of the sample (g), t denotes the reaction time (h).

### 2.8. Measurement of Biomass, R/S, SRL and SRA

The aboveground organs, taproots, and fine roots of *G. uralensis* were individually packed into paper bags and dried in an oven at 70 °C to constant weight, and their biomass was determined separately. Specific fine root length (SRL) and specific fine root surface area (SRA) were calculated using the following formulas:SRL (m/g) = FRL/M_fine roots_(3)
SRA (cm^2^/g) = FRA/M_fine roots_(4)
FRL denotes fine root length (m), FRA denotes fine root surface area (cm^2^), and M_fine roots_ denotes fine root biomass (g).

Root–shoot ratio (R/S) was calculated based on the ratio of the biomass of the underground organs (taproot biomass + fine root biomass) to the aboveground biomass (stems biomass + leaves biomass) using the following formula:R/S = M_underground organs_/M_aboveground organs_(5)
M_underground organs_ denotes underground biomass (g), M_aboveground organs_ denotes aboveground biomass (g).

Subsequently, the solid samples of *G. uralensis* roots were completely homogenized by grinding them to a fine powder with a grinder (ISO-9001, Guangsha Trade Co., Ltd., Shanghai, China) and passed through an 80-mesh nylon sieve.

### 2.9. Measurement of the Concentration of Medicinal Components and Phosphorus

To determine the concentration of the 3 medicinal components (glycyrrhizic acid, glycyrrhizin, and liquiritigenin) in the taproots of each treatment, the ultra-high-performance liquid chromatography–tandem mass spectrometry (UHPLC–MS/MS) method (Agilent Technologies Inc., Santa Clara, CA, USA) was applied. The method of the standard curve, UHPLC, and MS conditions referenced Chen [39]. The specific steps were as follows:

① Standard curve preparation: 2.0 mg of glycyrrhizic acid, glycyrrhizin, and liquiritigenin reference substance were weighed accurately and placed in a 10 mL volumetric flask, respectively, dissolved with methanol (chromatographically pure) to obtain the original solutions at the concentration of 200 μg·mL^−1^. The three original solutions were prepared as 1, 10, 50, 100, 250, 500, 1000 ng·mL^−1^ diluent solutions in methanol, respectively. The standard curves were plotted with the quantified ion peak area as the ‘Y’ coordinate and the concentration (ng·mL^−1^) as the ‘X’ coordinate. The results showed that within the range of 1–1000 ng/mL, the concentration of the 3 standard compounds had a good linear relationship with the peak area. The ion pairs, fragmentation voltages, and collision energies used for quantitative analysis are shown in Table 1.

② Sample extraction: 1.00 g sieved powder of each sample was accurately weighed and transferred into individual test tubes, then each test tube was filled with 10 mL methanol and extracted by ultrasonic extraction at room temperature for 2 h. Then, we centrifuged them at 12000 r·min^−1^ for 30 min and the supernatant was passed through a 0.25 μm filter membrane.

③ Assay: mobile phase: 0.1% formic acid–water (A) and acetonitrile (B) solution. Isocratic elution: 0–3.0 min, 20–98% B; 3.0–4.5 min, 98% B; 4.5–5.0 min, 98–20% B; 5.0–7.0 min, 20% B. Chromatographic conditions: The column was a Waters ACQUITY UPLC BEH C18 column (50 mm × 2.1 mm, 1.7 μm) (Model: 176000863, Waters Corporation Shanghai, Shanghai, China) with a flow rate of 0.3 mL.min^−1^, column temperature of 30 °C, and injection volume of 1 μL. Mass spectrometry conditions: The ion source was an electrospray ionization source (ESI) and multiple reaction detection mode (MRM) was used for the determination of content; desolvation gas temperature: 450 °C; ion source temperature: 150 °C; desolvation gas flow rate: 800 L·h^−1^; cone gas flow rate: 150 L·h^−1^; the capillary voltage was 2300 V. Each treatment was repeated three times, the mean value was calculated, and the content of each substance was calculated according to the regression equation.

The extraction method, standard curve, and the determination of total flavonoids were referenced from He et al. [40], and the concentration of the total flavonoids was determined using a UV–visible spectrophotometer (UV-1900, Shimadzu Corporation, Shanghai, China) at 490 nm. The concentration of the total flavonoids had a good linear relationship with the peak area within the range of 1–1000 ng/mL (Regression equation: Y = 0.026X − 0.056, *R*^2^: 0.9987).

The vanadium–molybdenum yellow colorimetric method [41] was used to determine the concentration of phosphorus (PC, mg/g) of the aboveground and underground organs of *G. uralensis*. The concentrations were determined using a UV–visible spectrophotometer (UV-1900, Shimadzu Corporation, Shanghai, China) at 450 nm (Regression equation: Y = 0.0068X + 0.0015, *R*^2^: 0.9998).

### 2.10. Determination of Lanthanum Concentration in the Taproots

Inductively coupled plasma mass spectrometry (ICP-MS) was used to determine the concentration of lanthanum in *G. uralensis* taproots, and the concentrations were quantified using 0–1000 ppb lanthanum standards (GSBG62047-90). The determination method was referenced from the China National Standard (GB/5009.94-2012) [42]. Initially, 0.200 g powdered root of each sample was accurately weighed and transferred into individual TFMs. Each TFM was filled with 5 mL of HNO_3_ solution (65%, Adamas Reagent Co., Shanghai, China) and 2 mL of H_2_O_2_ solution (30%, Adamas Reagent Co., Shanghai, China) and maintained at 25 °C for 1 h. Then, the acid digestion step of the sample solution was carried out using a temperature-controlled microwave digestion system (EHD-24 Electric Heating Digestion Instrument, Donghang Scien-Tech Instrument Co., Ltd., Beijing, China), and the digestive procedure was as follows (Table 2). After that, we cooled these solutions down to room temperature and they were individually transferred to a 140 °C temperature-controlled electric heating plate to evaporate the sulfuric acid from the sample solution, and we continued the evaporating process until the solution was clear and only soybean-sized droplets remained. Then, they were transferred to 10 mL volumetric flasks and each volumetric flask was spiked with 2% HNO_3_ solution (65%, Adamas Reagent Co., Ltd., Shanghai, China). The concentration of lanthanide in the solutions was determined by ICP-MS (Agilent 7500c, Agilent Technologies, Inc., Santa Clara, CA, USA), the determination was repeated three times for each treatment, and the instrumental parameters of ICP-MS are shown in Table 3.

### 2.11. Data Processing

Statistical analysis of all experimental data was performed by IBM SPSS 19.0 (IBM Corp., Armonk, NY, USA) software. Plotting of all experimental data was carried out in Graphpad prism software (version 9.0, GraphPad Software, Inc., La Jolla, CA, USA), and all datasets were verified to be normally distributed using the D’Agostino and Pearson omnibus normality test. Differences in organ biomass, the root system’s morphological parameters, R/S, root activity, concentration of medicinal components, concentration of phosphorus of aboveground and underground organs, and concentration of taproot lanthanum among the different treatments were statistically analyzed using post-hoc analyses (Tukey’s HSD; *p* < 0.05) after two-way ANOVA. A structural equation model (SEM) was constructed by using the “sem” and “DiagrammeR” packages.

## 3. Results

### 3.1. Effects of Foliar Application of Lanthanum on the Root System Characteristics

Among the P-deficient conditions, the growth of the *G. uralensis* root system appeared to be significantly inhibited, resulting in a decrease in the number of lateral roots and an increase in the length of individual lateral roots. However, this inhibitory effect was mitigated by the foliar application of 0.2–0.4 mM LaCl_3_ (later in the text, it will be shortened to 0.2–0.4 mM LaCl_3_, and so on for other similar treatments). Moreover, the treatments also exhibited significant enhancements, not only under P-deficient conditions but also under P-sufficient conditions, as evidenced by a greater length and quantity of lateral and fine roots, as well as stronger taproots (Figure 1).

The TRL, RB, RA, and R/S of the CK group increased significantly with the increase in soil phosphorus supply (Table 4). Compared with the CK group under P-deficient conditions, the above indices of the CK group increased by 19.57%, 25.54%, 6.60%, and 26.36%, respectively, under P-sufficient conditions (Figure 2).

Concentrations of 0.2–0.4 mM LaCl_3_ significantly promoted the TRL, accumulation of RB, RA under P-deficient and P-sufficient conditions, as well as R/S in the groups under P-deficient conditions. 0.4 mM LaCl_3_ groups had the highest values for these four indices, among the treatment groups under the same P-supply conditions. Compared with the CK group under P-deficient conditions, the above indices of the 0.4 mM LaCl_3_ group increased by 29.82%, 34.43%, 12.99%, and 18.77%, respectively, under P-deficient conditions. Compared with the CK group under P-sufficient conditions, the above indices of the 0.4 mM LaCl_3_ group increased by 26.66%, 18.86%, 10.84%, and 7.32%, respectively, under P-sufficient conditions (Figure 2).

In addition, there was a significant interaction between phosphorus and LaCl_3_ application in the case of TRL, TRB, and RA (Table 4). Compared with the P-sufficient conditions, the intensity of the promotion of RB and R/S in the 0.4 mM LaCl_3_ group under P-deficient conditions was significantly higher, and the growth rate of the RB and R/S values increased by 45.23% and 60.99%, respectively (Figure 2). 

### 3.2. Effects of Foliar Application of Lanthanum on Morphological Parameters of Taproots

The TL and TD of the CK group increased significantly with the increase in soil phosphorus supply (Table 5). Compared with the CK group under P-deficient conditions, the above indices of the CK group increased by 7.14% and 11.32%, respectively, under P-sufficient conditions (Figure 3).

Concentrations of 0.2–0.4 mM LaCl_3_ significantly promoted the TL and TD under P-deficient and P-sufficient conditions. Moreover, the 0.4 mM LaCl_3_ groups had the highest values for these two indices among the treatment groups under the same P-supply conditions. Furthermore, compared with the TD, the intensity of the promotion of the TL in the 0.4 mM LaCl_3_ group was significantly higher. Compared with the CK group under P-deficient conditions, the TL and TD values of the 0.4 mM LaCl_3_ group increased by 27.87% and 9.78%, respectively, under P-deficient conditions. Compared with the CK group under P-sufficient conditions, the above indices of the 0.4 mM LaCl_3_ group increased by 24.84% and 10.65%, respectively, under P-sufficient conditions (Figure 3).

### 3.3. Effects of Foliar Application of Lanthanum Addition on Morphological Parameters of Fine Roots

The FRD of the CK group exhibited a notable rise with the increase in soil phosphorus supply, while the SRL of the CK group decreased significantly, and there was little to no effect seen on the SRA (Table 6). Compared with the CK group under P-deficient conditions, the above indices of the CK group under P-sufficient conditions exhibited a 7.14% rise and an 11.32% reduction, respectively (Figure 4).

Concentrations of 0.2–0.6 mM LaCl_3_ significantly promoted the FRD under P-deficient and P-sufficient conditions. Moreover, the 0.6 mM LaCl_3_ groups had the highest values among the treatment groups under the same P-supply conditions. Compared with the CK group, the FRD of the 0.6 mM LaCl_3_ groups under P-deficient and P-sufficient conditions exhibited a 14.26% and a 16.94% rise, respectively.

Concentrations of 0.2–0.6 mM LaCl_3_ significantly decreased the SRL and SRA under P-deficient and P-sufficient conditions. Moreover, the 0.6 mM LaCl_3_ groups had the lowest values among the treatment groups under the same P-supply conditions.

Compared with the CK group under P-deficient conditions, the SRL and SRA values of the 0.6 mM LaCl_3_ group decreased by 27.78% and 10.98%, respectively, under P-deficient conditions. Compared with the CK group under P-sufficient conditions, the above indices of the 0.6 mM LaCl_3_ group decreased by 22.87% and 11.69%, respectively, under P-sufficient conditions (Figure 4). There was a significant interaction between phosphorus and LaCl_3_ application in the case of the SRL (Table 6).

### 3.4. Effect of Foliar Application of Lanthanum on Phosphorus Uptake and Phosphorus Partitioning Parameters

The root APase activity of the CK group decreased significantly with the increase in soil phosphorus supply (Table 7). Compared with the CK group under P-deficient conditions, the APase activity of the CK group decreased by 40.80% under P-sufficient conditions (Figure 5).

Concentrations of 0.2–0.4 mM LaCl_3_ significantly promoted the root APase activity under P-deficient and P-sufficient conditions, and the 0.2 mM LaCl_3_ groups had the highest activity of root APase among the treatment groups under the same P-supply conditions. Compared with the CK group, the root APase activity of the 0.2 mM LaCl_3_ groups under P-deficient and P-sufficient conditions exhibited a 15.16% rise and a 12.48% rise, respectively.

P-deficiency resulted in a significant decrease in total phosphorus concentration in *G. uralensis*, and the phosphorus concentration in the aboveground and underground organs of the CK group exhibited a notable rise with the increase in soil phosphorus supply (Table 7). Compared with the CK group under P-deficient conditions, the phosphorus concentration in the aboveground and underground organs of the CK group under P-sufficient conditions increased by 445.23% and 286.42%, respectively (Figure 6).

Concentrations of 0.2–0.6 mM LaCl_3_ significantly promoted phosphorus concentration in the aboveground organs of *G. uralensis* under P-deficient and P-sufficient conditions, and 0.2–0.4 mM LaCl_3_ significantly promoted phosphorus concentration in the underground organs of *G. uralensis*, too. Moreover, the 0.4 mM LaCl_3_ groups had the highest phosphorus concentration in the aboveground and underground organs among the treatment groups under the same P-supply conditions. Compared with the CK group under P-deficient conditions, the phosphorus concentration in the aboveground and underground organs of the 0.4 mM LaCl_3_ group increased by 224% and 51.32%, respectively, under P-deficient conditions. Compared with the CK group under P-sufficient conditions, the above indices of the 0.4 mM LaCl_3_ group increased by 48.19% and 16.84%, respectively, under P-sufficient conditions (Figure 6). There was a significant interaction between phosphorus and LaCl_3_ application in the case of APase activity, phosphorus concentration in the aboveground and underground organs, and phosphorus allocation ratio (Table 7). It was observed that under P-deficient conditions, there was a reduction in phosphorus concentration in all parts of the plant. Nevertheless, this triggered a rise in the allocation of phosphorus to the root system. Furthermore, the foliar application of LaCl_3_ led to an even larger rise in the distribution of phosphorus to the aboveground organs compared to the root system in *G. uralensis*.

### 3.5. Effect of Foliar Application of Lanthanum on the Content of Concentration of Medicinal Active Ingredients

The concentrations of glycyrrhetinic acid, glycyrrhizin, total flavonoids, and liquirigenin in the taproots of *G. uralensis* in the CK group increased significantly with the increase in soil phosphorus supply (Table 8). Compared with the CK group under P-deficient conditions, the above indices of the CK group increased by 52.57%, 106.34%, 59.55%, and 54.47%, respectively, under P-sufficient conditions, and the most prominent effect was seen in the augmentation to the concentration of glycyrrhizin (Figure 7).

Concentrations of 0.2–0.4 mM LaCl_3_ significantly promoted the above indices under P-deficient and P-sufficient conditions, and the 0.4 mM LaCl_3_ groups had the highest values for these four indices among the treatment groups under the same P-supply conditions.

Compared with the CK group under P-deficient conditions, the above indices of the 0.4 mM LaCl_3_ group increased by 13.81%, 75.26%, 77.45%, and 78.47%, respectively, under P-deficient conditions. Compared with the CK group under P-sufficient conditions, the above indices of the 0.4 mM LaCl_3_ group increased by 13.86%, 16.75%, 46.86%, and 49.90%, respectively, under P-sufficient conditions (Figure 7).

Furthermore, there was a significant interaction between phosphorus and LaCl_3_ application in the case of glycyrrhetinic acid and glycyrrhizin (Table 8); while comparing the P-sufficient conditions to P-deficiency, it was shown that the foliar application of lanthanum considerably increased the concentration of glycyrrhetinic acid and glycyrrhizin in the taproots of *G. uralensis*.

### 3.6. Effect of Foliar Application of Lanthanum on Lanthanum Concentration in Taproots

The results indicated that the foliar application of lanthanum significantly increased the lanthanum content in *G. uralensis* roots, with the concentration of lanthanum in the taproots exhibiting a consistent upward trend as the concentration of foliar LaCl_3_ spraying increased. This pattern was observed in both P-deficient and P-sufficient soil conditions (Figure 8).

The concentration of lanthanum in the taproots of *G. uralensis* was less than 0.01 ppm across all LaCl_3_ treatment groups in this experiment, which was in accordance with the international safety standard [43,44].

### 3.7. Correlation Analysis and Structural Equations Modeling

A Pearson correlation analysis showed that the taproot length and maximum diameter, two of the most important indicators of the taproot yield of *G. uralensis*, were significantly positively correlated with the root biomass, total root length, fine root diameter, activity of roots, root acid phosphatase activity, root phosphorus concentration, and the foliar application of lanthanum; conversely, the specific fine root length, specific fine root surface area, and root–shoot ratio exhibited a significant negative correlation with these variables. The concentrations of medicinal active ingredients of *G. uralensis*, including glycyrrhetinic acid, glycyrrhizin, total flavonoids, and liquirigenin in the taproots of *G. uralensis*, were significantly positively correlated with the root biomass, taproot length, maximum diameter of taproots, total root length, fine root diameter, activity of roots, root phosphorus concentration, and the foliar application of lanthanum; conversely, the specific fine root length, specific fine root surface area, root acid phosphatase activity, and root–shoot ratio exhibited a significant negative correlation with these four variables (Figure 9).

The SEM indicated that the foliar application of lanthanum increased the yield and quality of *Glycyrrhiza uralensis* taproots under P-deficient and P-sufficient conditions in three ways (Figure 10). First, it enhanced the taproot properties, directly improving the root biomass, taproot length, taproot diameter, and medicinal active ingredients. Second, it indirectly improved the taproot yield and quality by effecting the fine roots’ morphology and the activity of enzymes, including increasing the fine root length and diameter and the activity of roots, and decreasing the specific fine root length and specific fine root surface area. Third, it indirectly promoted the taproot yield and quality by increasing the phosphorus uptake efficiency.

## 4. Discussion

### 4.1. Effects of Foliar Application of Lanthanum on the Root System’s Morphological Characteristics and Phosphorus Partitioning Parameters

The root system serves as the primary interface between the plant and its surrounding soil environment. Its branches exhibit morphological diversity and perform diverse functions [45]. For example, distal roots are categorized as first-order roots, roots that produce two or more first-order roots are classified as second-order, and so on. Various orders of roots serve distinct functions of plant underground organs; first-order and second-order roots (typically fine roots ≤ 2 mm in diameter) are primarily used for acquiring nutrients and water. These roots have short lifespans and are quickly regenerated. Additionally, they exhibit a high degree of plasticity in response to the concentration of available phosphorus in the soil during their growth. On the other hand, third-order and higher-grade roots are primarily involved in transportation, providing structural support, and storing organic matter. These roots have considerably longer lifespans compared to first- and second-order roots [46,47].

The growth characteristics of plants’ root systems exhibit several changes in response to P-deficient conditions. However, the specific pattern of these changes might differ among plants. Some plants rely on the regulation of physiological responses (roots conduct enzyme secretion, e.g., APase), and others rely on the regulation of root morphology. *G. uralensis*, which has evolved in dry-land conditions where phosphorus availability is often patchy, is expected to exhibit high plasticity [48]. Thus, we expect that, in addition to root morphological responses, enzymatic physiological responses are likely to occur simultaneously. In line with anticipated outcomes, the deprivation of phosphorus would adversely affect the comprehensive condition of the licorice root system’s growth. This would emerge as a decrease in the TRL, RB, number of roots, RA, and the development of root hairs. Consequently, the nutrient uptake capacity and nutrient status of the plant would be impacted. The application of LaCl_3_ has been found to mitigate the detrimental impacts of phosphorus deficiency on plants [49]. 

In situations when nutrients are limited, plants may demonstrate an adaptive response through the development of deeper taproots, which facilitates them to access deeper soil resources. However, this process requires a substantial allocation of resources and energy. In reality, plants often depend on the proliferation of flexible, fine roots to augment their ability to absorb soil moisture and mineral nutrients. However, this increase in fine root lengths often results in a compensated reduction in the diameter and length of the taproot.

Nevertheless, in the 0.2–0.4 mM LaCl_3_ treatment group under P-deficient conditions, the taproot growth of *G. uralensis* was considerably enhanced, as evidenced by the increased length and diameter of the taproots. Collectively, the application of the LaCl_3_ promoted the input of more plant resources into the higher-order roots, which was beneficial to the enhancement of the *G. uralensis* taproot yield.

For the lower-order roots of *G. uralensis*, the smaller diameter of fine roots allows for a more extensive exploration range with relatively low biomass investment, thereby enabling the acquisition of a greater amount of soil resources [50]. The specific root length (SRL m/g) and the specific surface area (SRA m^2^/g) serve as significant indicators for assessing the costs and benefits associated with root systems.

Previous studies on the patterns of the SRL and SRA of fine roots in response to exogenous additives have yielded conflicting results [51]. The results of this experiment showed that although there was a decrease in both the number and diameter of fine roots under P-deficiency conditions, the SRL and SRA instead increased significantly. This suggests that the root system actively adapts its morphology to optimize resource utilization. Specifically, the root system compensates for the increase in the total length of fine roots by decreasing the fine root diameter, resulting in a larger absorbing area and a wider range of root exploration. This adaptive response can be characterized as an “exploratory” root system.

The growth of fine roots of *G. uralensis* in the 0.2–0.4 mM LaCl_3_ treatment groups under P-deficiency conditions exhibited a tendency toward normal growth conditions and enhanced P uptake by increasing the number and the total length of fine roots. These findings align with previous research that highlights the role of LaCl_3_ in regulating root morphology [52]. Simultaneously, the application of LaCl_3_ resulted in a notable augmentation in the secretion of APase by the fine roots. These enzymes facilitate the breakdown of recalcitrant phosphorus compounds present in the soil, converting them into absorbable phosphate ions that can be absorbed and utilized by the roots. Consequently, this effectively improves the tolerance of *G. uralensis* to phosphorus stress. Previous studies have found that the application of La(III) to Arabidopsis (*Arabidopsis thaliana* (L.) Heynh.) leaves induces root endocytosis through the activation of the coordinated action of AtrbohD and jasmonic acid, and systemic endocytosis alters the accumulation of mineral elements in the roots, which in turn affects the growth of primary roots and the formation of lateral roots. Therefore, La-induced systemic endocytosis likely provides an explanation for REEs’ effect on phosphorus acquisition.

The results proved that the application of LaCl_3_ did significantly promote the concentration of phosphorus in *G. uralensis*. It was observed that under P-deficient conditions, the reduction in phosphorus concentration triggered a rise in the allocation of phosphorus to the root system. The foliar application of LaCl_3_ led to a rise in the distribution of phosphorus to the aboveground organs compared to the root system. It is postulated that the observed phenomenon could potentially be attributed to a trade-off in resource allocation within the plant, referred to as an “input–benefit” allocation trade-off [53]. Plants exhibit a strategic allocation of resources to different organs based on the availability of limiting resources, aiming to optimize overall plant performance. Specifically, when the root system effectively absorbs phosphorus from the soil and reaches a certain threshold, there is a corresponding increase in the allocation of phosphorus to the aboveground organs. 

### 4.2. Effects of Foliar Application of Lanthanum on the Quality Enhancement of G. uralensis Taproots

In the present study, the significant promotional effects of LaCl_3_ on the accumulation of secondary metabolites (glycyrrhetinic acid, glycyrrhizin, total flavonoids, and liquirigenin) in the taproots of *G. uralensis* were all verified in a strong dose-dependent manner. These findings align with previous research that highlights the role of LaCl_3_ in the formation and accumulation of plant secondary metabolites [54,55,56]. The findings of this research indicate that the increases in secondary metabolite concentrations are highly correlated with phosphorus concentrations in the roots. Therefore, it is hypothesized that the increases in the above secondary metabolite content are due to the improved phosphorus nutrition as influenced by LaCl_3_. To validate our hypothesis, we devised three indicators based on the criteria outlined by Baas and Kambers [57]. These indicators are as follows: 1. The concentration of lanthanum translocated to the root system exhibits a consistent upward trend as the concentration of the foliar application of LaCl_3_ increases, indicating the effective absorption of lanthanum by plants. 2. The application of foliar LaCl_3_ treatment in a concentration-dependent manner results in a considerable increase in the phosphorus content in the roots of *G. uralensis*. 3. The treatment group exhibiting the greatest accumulation concentration of phosphorus in the roots corresponds to the treatment group displaying the highest concentrations of secondary metabolites in the roots.

All these criteria were fulfilled in our study, and these results indicate that foliar application of lanthanum triggers the plant to regulate the uptake and accumulation of mineral elements in the roots, which subsequently encourages the buildup of pharmacologically active components in its roots, a perspective consistent with the latest research by Cheng et al. [58].

### 4.3. Safety Assessment of G. uralensis Taproots

REEs exhibit a wide distribution in several environmental compartments such as soil, water, and the atmosphere. Consequently, these elements can be found in microorganisms, the human body, plants, and animals. Lanthanum possesses a comparatively elevated atomic mass, and previous studies have shown that it predominantly accumulates in the roots after being absorbed by plants [25]. The root enrichment of lanthanum is attributed to a dual mechanism involving cell wall adsorption and phosphate precipitation. The former mechanism is particularly prominent in strongly acidic conditions, while the latter mechanism assumes a greater role as the pH level rises. Upon analyzing the lanthanum levels in the roots of *G. uralensis* using the established national standard monitoring method [42], it was ascertained that the observed content fell within the acceptable safety limits outlined by both the National Standard of the People’s Republic of China [43] and the International Codex Alimentarius [44]. Consequently, it can be concluded that the presence of lanthanum in these roots does not pose any discernible risk to human health.

## 5. Conclusions

In brief, this research clarified the effects of different concentrations of LaCl_3_ on root system morphology and changes in enzymes of *G. uralensis* under P-deficient and P-sufficient conditions. Furthermore, the study identified the optimal concentration of LaCl_3_ for application, which was found to substantially improve the yield and quality of Uralic licorice herbs while remaining within the safe application range. The foliar application of lanthanum can exert a positive influence on the development of both the taproots and fine roots of *G. uralensis*, thus broadening the root system’s exploration range (Figure 11). Additionally, this application significantly improves phosphorus uptake efficiency by stimulating root activity and APase activity, and regulates phosphorus partitioning and biomass allocation between the aboveground organs and root system. These combined effects amplify the root system’s capacity to absorb nutrients from the soil and enhance the individual’s nutritional status. Ultimately, this results in a substantial increase in both yield and quality.

## Figures and Tables

**Figure 1 plants-13-00474-f001:**
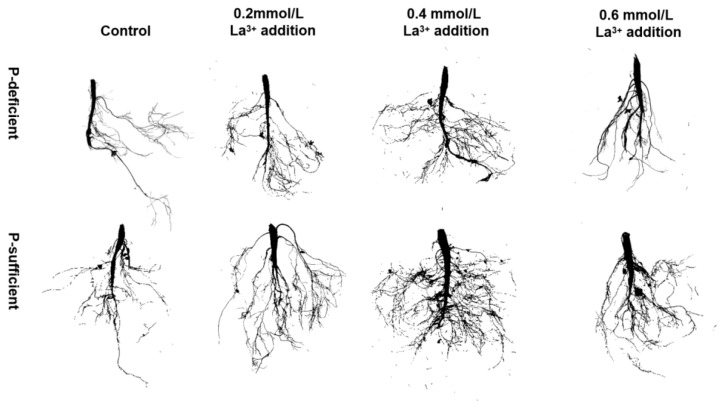
Diagram of root system scanning results of the *G. uralensis* treated with different concentrations of LaCl_3_ under P-deficient and P-sufficient conditions.

**Figure 2 plants-13-00474-f002:**
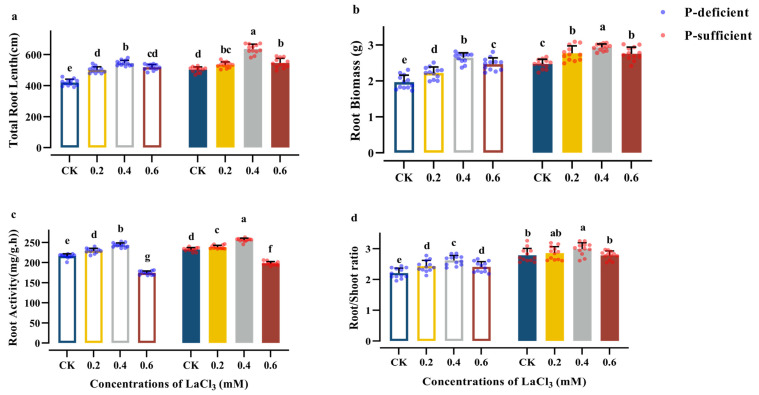
Effects of different concentrations of LaCl_3_ on TRL (**a**), TRB (**b**), RA (**c**), and R/S (**d**) of *G. uralensis* under P-deficient and P-sufficient conditions (mean ± SE). Different lowercase letters indicate significant differences in root system characteristics of *G. uralensis* among treatments (*p* < 0.05).

**Figure 3 plants-13-00474-f003:**
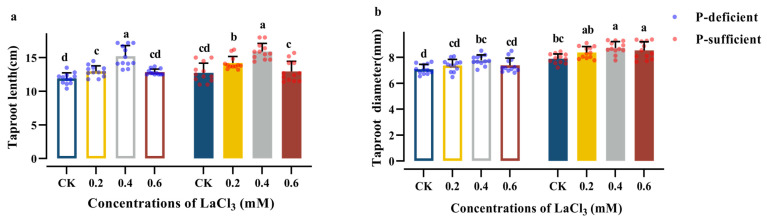
Effects of different concentrations of LaCl_3_ on TL (**a**) and TD (**b**) of *G. uralensis* under P-deficient and P-sufficient conditions (mean ± SE). Different lowercase letters indicate significant differences in morphological parameters of *G. uralensis* taproots among treatments (*p* < 0.05).

**Figure 4 plants-13-00474-f004:**
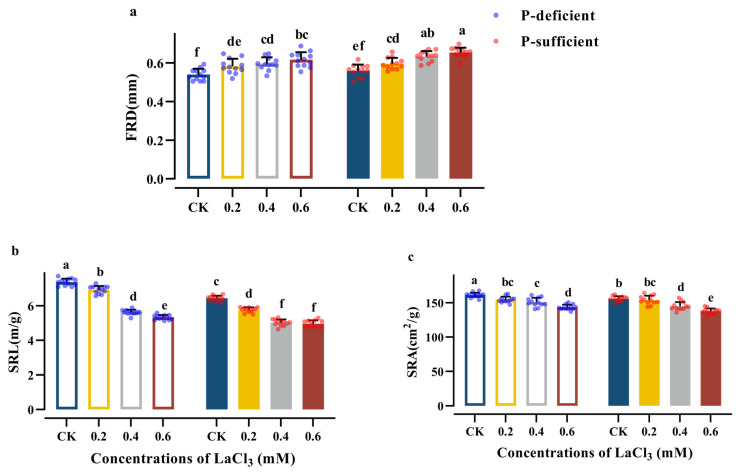
Effects of different concentrations of LaCl_3_ on FRD (**a**), SRL (**b**), and SRA (**c**) of *G. uralensis* under P-deficient and P-sufficient conditions (mean ± SE). Different lowercase letters indicate significant differences in morphological parameters of fine roots among treatments (*p* < 0.05).

**Figure 5 plants-13-00474-f005:**
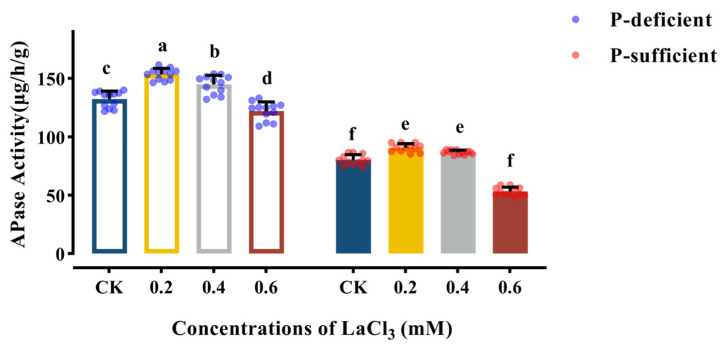
Effects of different concentrations of LaCl_3_ on root APase activity of *G. uralensis* under P-deficient and P-sufficient conditions (mean ± SE). Different lowercase letters indicate significant differences in APase activity among treatments (*p* < 0.05).

**Figure 6 plants-13-00474-f006:**
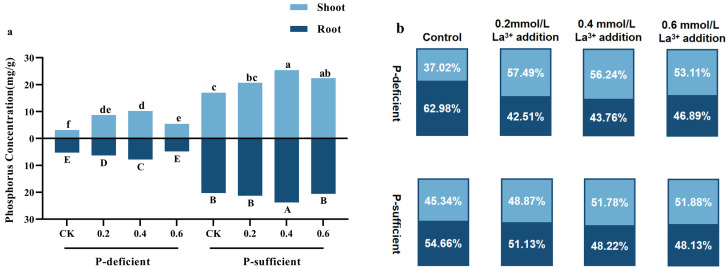
Effects of different concentrations of LaCl_3_ on phosphorus concentration (**a**) and phosphorus allocation ratio (**b**) in aboveground and underground organs of *G. uralensis* under P-deficient and P-sufficient conditions. Different lowercase and capital letters indicate significant differences of phosphorus concentration among treatments in aboveground and underground organs, respectively.

**Figure 7 plants-13-00474-f007:**
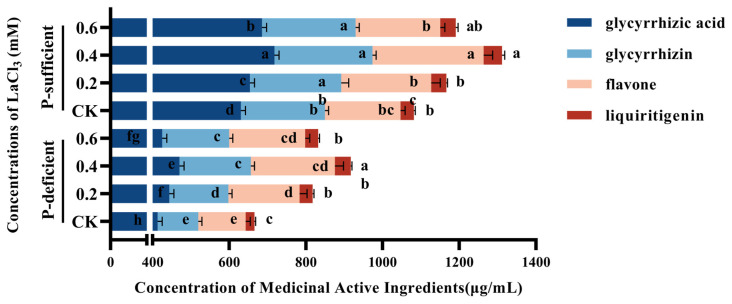
Effects of different concentrations of LaCl_3_ on concentrations of medicinal active ingredients of *G. uralensis* taproots under P-deficient and P-sufficient conditions (mean ± SE). Different lowercase letters indicate significant differences in concentration of medicinal active ingredients in *G. uralensis* taproots among treatments (*p* < 0.05).

**Figure 8 plants-13-00474-f008:**
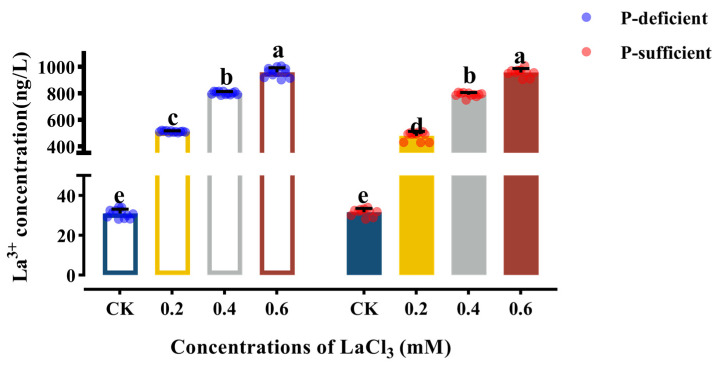
Effects of different concentrations of LaCl_3_ on taproot Lanthanum concentration of *G. uralensis* under P-deficient and P-sufficient conditions (mean ± SE). Different lowercase letters indicate significant differences in concentration of lanthanum in *G. uralensis* taproots among treatments (*p* < 0.05).

**Figure 9 plants-13-00474-f009:**
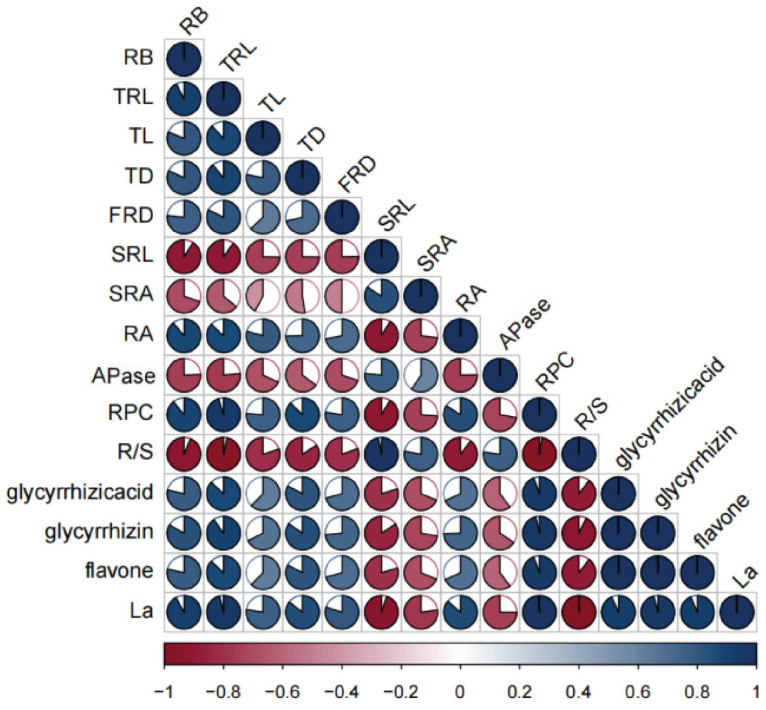
Correlations between foliar application of LaCl_3_ and the properties of fine roots and taproots under P-deficient and P-sufficient conditions. In the correlation coefficient matrix, positive correlations are displayed in blue and negative correlations in red color; the darker the colors, the larger the sector areas, and the higher the correlations between two variables. RB, root biomass; TRL, total root length; TL, taproot length; TD, maximum diameter of taproots; FRD, fine root diameter; SRL, specific fine root length; SRA, specific fine root surface area; RA, activity of roots; APase, root acid phosphatase activity; RPC, root phosphorus concentration; R/S, root–shoot ratio; La, foliar application of lanthanum.

**Figure 10 plants-13-00474-f010:**
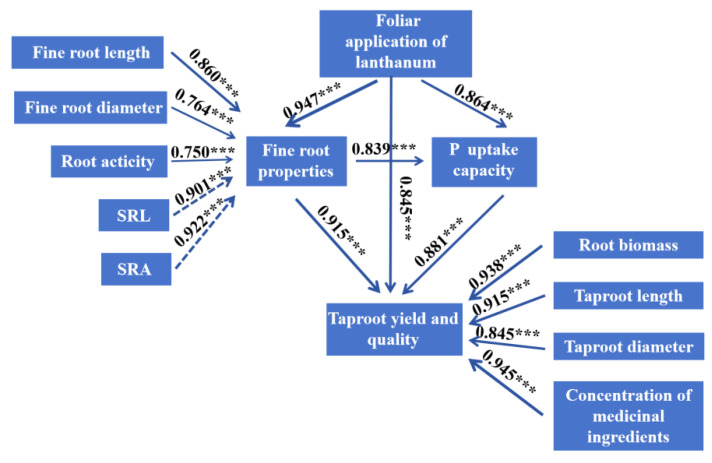
Structural equation modeling (SEM) of the direct or indirect effects of LaCl_3_ on the yield and quality of *Glycyrrhiza uralensis* taproots under P-deficient and P-sufficient conditions. (GFI = 0.72; CFI = 0.81; RMR = 0.06; *p* < 0.05). The full lines indicate positive effects, dotted lines indicate negative effects, the values next to the lines are the normalized pathway coefficients. ns denotes *p* > 0.05; *** denotes *p* ≤ 0.001.

**Figure 11 plants-13-00474-f011:**
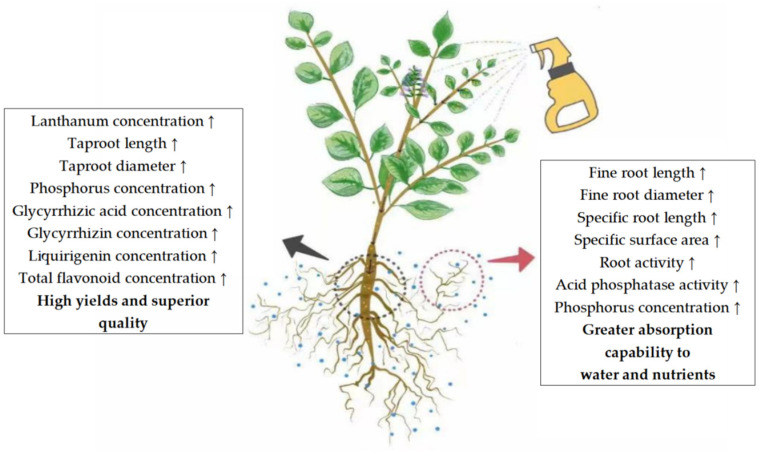
Pattern of the effects of foliar application of lanthanum on taproots and fine roots of *G. uralensis*.

**Table 1 plants-13-00474-t001:** Mass spectrometry conditions, regression equations, and correlation coefficients of 3 compounds.

Compounds	Parent Ion (*m*/*z*)	Daughter Ion (*m*/*z*)	Ionization Mode	Voltage (V)	Collisional Energy (V)	Retention Time (min)	Regression Equation	Correlation Coefficients *R*^2^	Linear Over (ng/mL)
Glycyrrhizic acid	821.2	350.9 *	−	62	42	2.48	Y = 87.2X − 3.76	0.9992	1.0–978.0
		113.0		62	56				
		134.9		2	18				
		409.2		100	46				
Glycyrrhizin	417.0	254.9 *	−	52	20	1.00	Y = 750.9X + 356.9	0.9999	1.0–992.3
		134.9		52	30				
Liquiritigenin	257.2	137.0 *	+	92	24	1.97	Y = 509.2X + 925.6	0.9980	0.9–962.5
		147.0		92	18				
		147.0		2	18				

Note: * stands for quota ion.

**Table 2 plants-13-00474-t002:** Procedure of digesting by microwave.

Move	Control Temperature (°C)	Temperature Rise Time (min)	Constant Temperature Time (min)
1	120	5	5
2	150	5	10
3	180	5	15

**Table 3 plants-13-00474-t003:** Instrument Parameters of ICP-MS.

Parameters	Numerical Value
RF power	1300 W
Sampling depth	6.8 mm
Auxiliary gas, plasma gas, and carrier gas	Nitrogen
Flow rates of auxiliary gas, plasma gas, and carrier gas	1.0, 16.0, 1.17 L/min
Atomizer type	Concentric nebulizer
Fog chamber temperature	2 °C
Sampling cone vs. interception cone type	Nickel cone
Scanning method	Peak-hopping mode
Collection points	3
Number of repetitions	3 times
Detector voltage	−12 V

**Table 4 plants-13-00474-t004:** Two-way ANOVA of the response of foliar application of lanthanum to indexes of *G. uralensis* root system characteristics under P-deficient and P-sufficient conditions.

Variables	Phosphorus		LaCl_3_		Phosphorus × LaCl_3_	
	*F*	Sig.	*F*	Sig.	*F*	Sig.
TRL	158.49	0.00	133.44	0.00	12.93	0.00
TRB	147.10	0.00	50.99	0.00	4.09	0.01
RA	228.34	0.00	705.24	0.00	8.96	0.00
R/S	135.01	0.01	11.71	0.06	1.73	0.17

**Table 5 plants-13-00474-t005:** Two-way ANOVA of the response of foliar application of lanthanum to indexes of morphological parameters of *G. uralensis* taproots under P-deficient and P-sufficient conditions.

Variables	Phosphorus		LaCl_3_		Phosphorus × LaCl_3_	
	*F*	Sig.	*F*	Sig.	*F*	Sig.
TL	9.17	0.00	35.66	0.00	1.0	0.4
TD	99.12	0.00	10.39	0.00	0.52	0.67

**Table 6 plants-13-00474-t006:** Two-way ANOVA of the response of foliar application of lanthanum to indexes of morphological parameters of *G. uralensis* fine roots under P-deficient and P-sufficient conditions.

Variables	Phosphorus		LaCl_3_		Phosphorus × LaCl_3_	
	*F*	Sig.	*F*	Sig.	*F*	Sig.
FRD	16.85	0.01	31.95	0.00	0.90	0.44
SRL	460.85	0.00	555.03	0.00	24.62	0.00
SRA	18.16	0.00	58.85	0.00	1.50	0.22

**Table 7 plants-13-00474-t007:** Two-way ANOVA of the response of foliar application of lanthanum to indexes of phosphorus uptake and phosphorus partitioning parameters of *G. uralensis* under P-deficient and P-sufficient conditions.

Variables	Phosphorus		LaCl_3_		Phosphorus × LaCl_3_	
	*F*	Sig.	*F*	Sig.	*F*	Sig.
APase	298.07	0.00	179.91	0.00	10.24	0.00
P concentration	560.23	0.00	205.03	0.00	14.60	0.00
P allocation ratio	299.16	0.00	358.85	0.00	26.09	0.00

**Table 8 plants-13-00474-t008:** Two-way ANOVA of the response of foliar application of lanthanum to indexes of concentration of medicinal active ingredients of *G. uralensis* taproots under P-deficient and P-sufficient conditions.

Variables	Phosphorus		LaCl_3_		Phosphorus × LaCl_3_	
	*F*	Sig.	*F*	Sig.	*F*	Sig.
glycyrrhizic acid	808.80	0.00	37.55	0.00	6.08	0.01
glycyrrhizin	335.05	0.00	30.21	0.00	4.93	0.01
flavone	54.06	0.00	27.658	0.00	2.62	0.09
liquiritigenin	19.31	0.00	13.31	0.00	0.76	0.53

## Data Availability

Data are contained within the article.

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
