# Peer review of "Lanthanum Significantly Contributes to the Growth of the Fine Roots’ Morphology and Phosphorus Uptake Efficiency by Increasing the Yield and Quality of Glycyrrhiza uralensis Taproots"

_plants, 2024, doi:10.3390/plants13040474_

Round 1

Reviewer 1 Report

Comments and Suggestions for Authors

The manuscript titled "Lanthanum significantly contributes to the growth of the fine roots' morphology and phosphorus uptake efficiency by increasing the yield and quality of Glycyrrhiza uralensis taproots" is interesting, novel and providing good information on the use of Lanthanum in plants, which scientifically brings something new to humanity. However, in my opinion, an adequate statistical analysis of the data obtained in the research was not carried out and therefore, the wording of the results must be substantially improved to make it more understandable for readers. Below I highlight some more specific details that could help improve the quality of the manuscript:

L144: You must separate the units of measurement from the number. for example: '25 °C' Check the entire document (L198, 199...).

L152: "phosphorus concentrations were measured..." please explain how those measurements were carried out.

L154: "and 6 G." When numbers less than 10 are referred to in the writing of a text, they must be written in letters. Please change the 6 to six. The same applies to "4 plants" on line 158.

L197: "0.300g root’s tissue of each sample was accurately weighed" is it important to explain which part of the root was whole? If the different parts of the root were mixed (taproots or fine roots), were they chopped, or if only a random piece was cut, etc. They must describe the procedure they followed to subsequently weigh the 300 mg.

L238: "UHPLC and MS conditions reference to Fu et al.[33]." In this reference, they used an Agilent HPLC/Q-TOF/MS equipment. You must describe the brand and model of the equipment used and the conditions, since Fu et al. They did not determine your three compounds (glycyrrhizic acid, glycyrrhizin and liquiritigenin).

L239-240: "the concentration of the 5 standard compounds" which 5? They only report 3. Please verify and correct.

L285-289: I consider that the statistical analysis of the data is poorly applied. Having done an independent one-way ANOVA for each phosphorus level considering only LaCl3 concentrations as a source of variation, leaves a lot of information up in the air. I recommend carrying out the data analysis under a completely randomized design with a 2X4 factorial arrangement where both the two levels of phosphorus and the 4 concentrations of LaCl3 are being evaluated, plus the effect of the interaction of both. This information would be very interesting and enriching the results.

L310-311: "TRL, RB, RA and R/S of the CK group increased significantly with the increase of soil phosphorus supply." How can you ensure this, if you did not do a statistical analysis comparing the effect of phosphorus supply? That is why it is important to analyze the results under a design with factorial arrangement as I suggest.

L311-313: These percentages cannot be used to compare whether there are significant statistical differences between the treatments, unless they have done one-way statistical analysis for each level of LaCl3. Which is not correct. Furthermore, according to the description in Figure 2, the blue dots are for P-sufficient and the red dots are for P-deficient; therefore, the highest values would correspond to P-deficient. Contrary to what they claim. Please verify that what is described in the figure matches the wording.

L330-331: Similar case, how can they ensure a significant increase in the level of phosphorus in the soil, if they did not do the statistical analysis comparing only the level of phosphorus, and if they did, where do these significant differences indicate? Check this condition throughout the document.

L335-336: "0.4 mM LaCl3 groups had the highest values for these two indices, among the treatment groups under the same P-supply condition." This is not true, since for TD there were no significant differences between 0.2 with 0.4 and 0.6 in each P-supply group. Please check the results in Figure 3 and correct the wording.

Figure 6: It is a scientific error to make a comparison of means between two different sources of variation (phosphorus and LaCl3) considering it as if it were a single treatment. They are two different effects that are influencing each other. Please correct the experimental design for a factorial one.

L452-453: "positive correlations are displayed in red and negative correlations in blue color" The lower bar of Figure 9 indicates negative correlations in red and positive correlations in blue. Please check and correct the slash or wording.

L489-502: All this information is background information that should be included in the introduction section, not in discussions. Please change the section.

L581, L594: "Our findings indicate" every scientific manuscript must be written in the third person. Please change the wording. For example: 'the findings of this research...'

L586: "by Baas and Lambers[54]" in the references section, this quote corresponds to Baas and Kuiper, please correct it.

Author Response

Dear Editor,

I hope this email finds you well.
Thank you very much for taking the time to review this manuscript. We express our sincere apologies for the mistake and the inconvenience it has caused you. We particularly appreciate your kind words and reminder. I have revised the manuscript according to the referees' comments. Please find the detailed corresponding corrections highlighted in the re-submitted files(Please see the attachment).

We deeply appreciate your consideration of our manuscript. If you have any queries, please don’t hesitate to contact me anytime. Thank you and best regards.

 Yours sincerely,

 Ma Miao

Reviewer 2 Report

Comments and Suggestions for Authors

The authors performed simple pot experiment with two doses of
phosphorus fertilization and four levels of lanthanum addition
to check the influence of such treatments on root parameters
and chemical composition of Glycyrrhiza uralensis.
The results are valuable and have practical meaning in terms of
better usage of phosphorus, however manuscript needs
improvement. Statistical methods needs revision.
There are also many errors – both in description of the results
which are presented in the figures as well as editorial.
Details are below

Line 13 – 18 – this sentence has repeated information „the effects of different concentrations of LaCl3 on …” and „..in response to the supply of varying concentrations of LaCl3 on G. uralensis,,”

There are many punctuation errors – lack of spaces – for example lines 82 – 84, but also in the rest of the manuscript. Moreover, way of referencing in the text is not according to the publisher requirements, as well as list of references.

 Line 87 – You have mentioned that „..few research….”, so give these references here.

Line 109 -  should be from Thermo….

Line 137 – check please this available phosphorus content - was it really 5.72 g/kg? Especially, that after adding fertilizer soil available phosphorus concentrations were measured to be 51.2 and 23.2 mg/kg, respectively. So, how is this possible?

Line 164 - P-sufficient and P-deficient

Line 164 – 170 – correct this part, so it is not as instruction.

 Line 289 – two -way Anova would be better because you have two factors: phospohorus fertilization and lanthanum addition

Figure 2 – better will be to mark differences with different letters (as it was done in Figure 5). In such way of showing differences as you have done we do not know if 0,2 is different from 0,6 or not. Moreover, was this Anova performed separatelly for P-deficient and P-sufficient treatments? From Figure 2 I supose that it is like that and again better will be to perform two way Anova – You will see if there are significant interactions between P and lanthanum.

Line 315 – „0.2-0.4 mM LaCl3 significantly promoted the…..as well as R/S in the groups under P-deficient condition…”- According to Figure 2 that is not true – differences in the case of P – deficient group (red dots) are not significant

Line 323 – „…promotion for RB and R/S was significantly higher, …” – again how do you know that if Anova were performed separatelly?

Lines 330 -332 – From Figure 3 wee can see that CK group in P- deficient treatment (red dots) has higher TD than in P – sufficient (blue dots) so it is completelly opposite than you wrote

Figure 4 – why you change the color of dots here – this time blue dots are deficient and red dots are sufficient? It is misleading for readers. I suggest that you describe in the Figure captions exactly which traetments are P-deficient and which are P-sufficient or mark it as it is in Figure 5.

Lines 454 – 458 – this should be part of Figure description.

Lines 452 – 453 – In Figure 9 there is exactly opposite – red colour is for negative correlations and blue is for positive.

Comments on the Quality of English Language

English needs revision

Author Response

(The authors gave the same response as above.)

Round 2

Reviewer 1 Report

Comments and Suggestions for Authors

The manuscript entitled "Lanthanum significantly contributes to the growth of the fine roots' morphology and phosphorus uptake efficiency by increasing the yield and quality of Glycyrrhiza uralensis taproots" was considerably improved with the corrections made and the results became more understandable with the change in the statistical analysis ; However, there are still mostly spelling errors that need to be checked and corrected.

I point out some details that I observe in this new version and that must be addressed:

L208: "0.300g" must put a space between the number and the unit of measurement. Check this recurring error throughout the document. L274

L223: "McLachlan et al.[35]." The word et al must be in italics and a space between the period and the reference. Check the entire document L257. L261...

L251-252: "UHPLC and MS condition reference to Chen.[36]." Same observation as the last review; They did change the reference, but for a master's thesis that is difficult to consult, so I request that they describe all the characteristics of the equipment used and conditions of the analysis. Since all methodologies must be reproducible.

L318: "(P<0.05)" the probability symbol must be in lower case, please review the entire document and correct. example: '(p<0.05)'.

Author Response

Thank you for your comments about our manuscript. Those comments are precious and helpful for revising and improving our paper and the important guiding significance to our following works. We have made corresponding corrections point-by-point by highlighting them in green in the re-submitted files.
Question 1. Line 208 and Line 247 have been revised.
Question 2. Line 223, Line 257 and Line 261 have been revised.
Question 3. Line 251-252: "UHPLC and MS condition reference to Chen.[36]." I’ve described all the characteristics of the equipment used and the conditions of the analysis.
Question 4. Line 318 has been revised.

Reviewer 2 Report

Comments and Suggestions for Authors

The manuscript is corrected sufficiently. There are however still some editorial errors, for example check latin name of G. uralensis - sometimes it is written in normal font, while it should be in italics - line 453.

lines 395 - 396 - this sentence should be rather "There was significant interaction between phosphorus and LaCl3 application in the case of SRL"

 lines 343 - 344 - similarly as above, chcek also in other places in the manuscript

There should be always space between word and bracket (for example when you are mentioning tables or figures) - correct in the whole manuscript, as well as between word and references.

Phosphorus - shouldn't be written with capital letter when in the middle of the sentence

Comments on the Quality of English Language

English needs revisions

Author Response

Thank you for your comments about our manuscript. Those comments are precious and helpful for revising and improving our paper and the important guiding significance to our following works. We have made corresponding corrections point-by-point by highlighting them in green in the re-submitted files.
Question 1. The Latin name of G. uralensis has been revised in italics.
Question 2. Lines 395 - 396 and similar places in the manuscript have been revised.
Question 3. The space between the word and bracket and similar places in the manuscript has been revised.
Question 4. “Phosphorus” has been revised.